# HIV-1 Vif disrupts phosphatase feedback regulation at the kinetochore, leading to a pronounced pseudo-metaphase arrest

Dhaval Ghone[1,2†], Edward L Evans[1,3,4†‡], Madison Bandini[1,3,4], Kaelyn G Stephenson[1], Nathan M Sherer[1,4,5*], Aussie Suzuki[1,5*]

[1]McArdle Laboratory for Cancer Research, Department of Oncology, University of Wisconsin-Madison, Madison, United States; [2]Biophysics Graduate Program, University of Wisconsin-Madison, Madison, United States; [3]Cancer Biology Graduate Program, University of Wisconsin-Madison, Madison, United States; [4]Institute for Molecular Virology, University of Wisconsin-Madison, Madison, United States; [5]Carbone Comprehensive Cancer Center, University of Wisconsin-Madison, Madison, United States

*For correspondence:
nsherer@wisc.edu (NMS);
aussie.suzuki@wisc.edu (AS)

†These authors contributed equally to this work

Present address: ‡Laboratory for Optical and Computational Instrumentation, University of Wisconsin-Madison, Madison, United States

Competing interest: The authors declare that no competing interests exist.

## eLife Assessment

This study provides a **convincing** explanation for why HIV-1 Vif causes a qualitatively different cell cycle arrest to its accessory gene counterpart Vpr. The authors use elegant time-dependent microscopy reporter assays in immortalized tumor cell models to show that HIV-1 Vif causes a pseudo-metaphase arrest rather than a G2 arrest. The metaphase arrest correlates with dysregulation of the kinetochore that could be explained by the loss of phosphatase functions that determine chromosome-microtubule interactions. These **valuable** findings lay the groundwork for additional studies examining the mechanisms and consequences of this Vif-dependent phenotype in the viral life cycle and in primary cells more relevant to HIV-1 pathogenesis.

**Abstract** Virion Infectivity Factor (Vif) of the Human Immunodeficiency Virus type 1 (HIV-1) targets and degrades cellular APOBEC3 proteins, key regulators of intrinsic and innate antiretroviral immune responses, thereby facilitating HIV-1 infection. While Vif's role in degrading APOBEC3G is well-studied, Vif is also known to cause cell cycle arrest, but the detailed nature of Vif's effects on the cell cycle has yet to be delineated. In this study, we employed high-temporal resolution single-cell live imaging and super-resolution microscopy to monitor individual cells during Vif-induced cell cycle arrest. Our findings reveal that Vif does not affect the G2/M boundary as previously thought. Instead, Vif triggers a unique and robust pseudo-metaphase arrest, distinct from the mild prometaphase arrest induced by Vpr. During this arrest, chromosomes align properly and form the metaphase plate, but later lose alignment, resulting in polar chromosomes. Notably, Vif, unlike Vpr, significantly reduces the levels of both Protein Phosphatase 1 (PP1) and 2 A (PP2A) at kinetochores, which regulate chromosome-microtubule interactions. These results unveil a novel role for Vif in kinetochore regulation that governs the spatial organization of chromosomes during mitosis.

## Introduction

The human immunodeficiency virus type 1 (HIV-1) weakens the immune system by depleting CD4 +T cells, eventually causing the Acquired Immunodeficiency Syndrome (AIDS; *Deeks et al.,*

2015; Swanstrom and Coffin, 2012). Consequently, individuals infected with HIV-1 have an increased susceptibility to specific cancers and other health complications (Cohen et al., 2016; Grulich et al., 2007; Hernández-Ramírez et al., 2017; Parkin, 2006). After HIV-1 enters a host cell, its RNA genome undergoes reverse transcription to form double-stranded DNA, followed by integration of the DNA provirus into the host's genome. Using the host's transcriptional machinery, HIV-1 transcribes its genome into spliced, partially spliced and completely unspliced viral mRNAs, facilitating viral gene expression and infectious virion production (Freed, 2015; Karn and Stoltzfus, 2012). While the mechanisms underlying CD4 +T cell depletion during HIV-1 infection remain an active area of research, evidence suggests that both direct cytopathic effects of HIV-1 and chronic hyperactivation of the immune system contribute significantly. These processes drive apoptosis and induce pyroptosis in CD4 +T cells, leading to their progressive loss (Doitsh and Greene, 2016; Vidya Vijayan et al., 2017).

HIV-1 encodes four accessory viral proteins (Vif, Vpr, Vpu, and Nef) that are nonessential for virus replication in some ex vivo cell culture systems (Gabuzda et al., 1992) but play crucial immunomodulatory roles in vivo (Malim and Emerman, 2008). The primary role of Vif (Virion Infectivity Factor) is to facilitate the proteasomal degradation of APOBEC3 (A3) family of cytidine deaminases (e.g. A3F, A3G, and A3H). A3 proteins introduce deleterious mutations into the HIV-1 genome by deaminating cytosine residues in the viral single-stranded DNA during reverse-transcription, converting them to uracil (Chiu and Greene, 2009; Okada and Iwatani, 2016). Vif orchestrates A3 protein degradation by recruiting an E3 ubiquitin ligase complex (Conticello et al., 2003; Marin et al., 2003; Sheehy et al., 2003; Stopak et al., 2003; Yu et al., 2003). This degradation prevents A3 proteins from being incorporated into budding viral particles, ensuring that the progeny virions remain infectious.

Independently of its primary role of A3 protein degradation, several studies have shown Vif to induce cell cycle arrest and cell death in CD4 +T cells and several other cell types (DeHart et al., 2008; Du et al., 2019; Greenwood et al., 2016; Marelli et al., 2020; Nagata et al., 2020; Sakai et al., 2006; Salamango et al., 2019; Zhao et al., 2015). However, the molecular mechanisms that underpin these effects remain unclear. An earlier study suggested that p53, a major tumor suppressor protein, is required for Vif-induced G2/M cell cycle arrest (Izumi et al., 2010). Other studies demonstrated a relationship between Vif and Cyclin F (Augustine et al., 2017), a non-canonical cyclin critical for late S- and G2-phase progression (Clijsters et al., 2019; Enrico et al., 2021), as well as between Vif and Cdk1 and Cyclin B1, which are essential for the transition into and out of mitosis (Sakai et al., 2011). More recently, several studies have shown that Vif's cell cycle arrest activity correlates with the loss of B56 proteins, which are regulatory subunits of protein phosphatase 2A (PP2A; DeHart et al., 2008; Du et al., 2019; Greenwood et al., 2016; Marelli et al., 2020; Nagata et al., 2020; Salamango et al., 2019; Salamango et al., 2020; Zhao et al., 2015). The PP2A-B56 complex is known to play a critical role in various key processes during G2 and mitosis (Foley et al., 2011; Lee et al., 2017; Schuhmacher et al., 2019).

These prior studies have predominantly employed flow cytometry-based techniques to measure cell cycle phase population densities. However, flow cytometry has limitations its ability to differentiate between late S, G2, and M phases, because it categorizes cell cycle phases solely based on relative DNA content. Accordingly, in this study we prioritized high-temporal resolution single-cell live imaging that would allow us to directly observe the disruptions of the cell cycle triggered by Vif expression. We demonstrate that Vif induces a highly unique and robust pseudo-metaphase arrest, irrespective of the cell line tested or its p53 status. Additionally, we found that Vpr unexpectedly induces a distinct mitotic delay, clearly different from the pseudo-metaphase arrest caused by Vif. Vif, but not Vpr, disrupts the localizations of PP2A-B56 at the kinetochores during prometaphase, leading to a slight yet significant delay in the alignment of chromosome at metaphase. This disruption results in reduced localization of the Astrin-SKAP-PP1 complex at kinetochores, causing improper kinetochore-microtubule binding affinity due to increased phosphorylation of a microtubule binding protein, Hec1, at the kinetochores. These effects result in unbalanced forces between sister chromatids, resulting in misaligned chromosomes and abnormal chromosomal movements. These insights provide a deeper understanding of Vif's impact on the regulation of the host cell cycle, a conserved feature of Vif that may have potential relevance to HIV-1 pathogenesis in vivo.

# Results
## Vif and Vpr induce distinct forms of mitotic arrest

Previous research demonstrated that both Vif and Vpr expression causes cell cycle arrest and cytotoxicity in CD4 +T cells and as well as many cancer cell lines (*Augustine et al., 2017*; *Emerman, 1996*; *Evans et al., 2018*; *Nagata et al., 2020*; *Sakai et al., 2011*; *Sakai et al., 2006*; *Salamango et al., 2019*; *Salamango et al., 2020*; *Wang et al., 2011*; *Wang et al., 2008*). To investigate the nature of the cell cycle arrest induced by Vif, we employed high-temporal resolution live-cell imaging using the triple negative breast cancer Cal51 cell line. This cell line was chosen for several reasons; it has been engineered for precise cell cycle tracking through CRISPR-Cas9-mediated endogenous tagging of Histone H2B with mScarlet, allowing visualization of DNA, and Tubulin with mNeonGreen to enable monitoring the microtubule cytoskeleton (*Scribano et al., 2021*; *Figure 1A*). This approach minimizes the confounding effects of exogenous overexpression of fluorescently labeled Histone and Tubulin proteins on cell cycle progression. Moreover, Cal51 cells are well-suited for long-term live cell imaging assays due to their adherent growth properties, which facilitate extended single-cell monitoring. They also exhibit a stable, near-diploid karyotype and retain wild-type p53 expression, making them an ideal model for cell cycle research (*Lynch et al., 2022*). For these experiments, we infected cells with the NL4-3 strain HIV-1 reporter viruses expressing either Vif ('Vif'), Vpr ('Vpr'), a combination of both ('Vif +Vpr'), or a lack of both ('Control'; *Figure 1—figure supplement 1A*; *Evans et al., 2018*). Note that the NL4-3 strain encodes all known HIV-1 proteins and serves as a well-established model for studying HIV-1 biology (*Mustafa and Robinson, 1993*). These reporter viruses express cyan fluorescent protein (CFP) allowing us to identify infected cells using fluorescence microscopy. To focus our study on the effects of Vif on the cell cycle, these reporter viruses were modified not to express viral Env and Nef proteins, which are known to exhibit cytotoxicity (*Elder et al., 2002*; *Emerman, 1996*; *Evans et al., 2018*).

We assessed the mitotic duration defined as the time between nuclear envelope breakdown (NEBD) and anaphase onset in CFP-positive cells using video microscopy. Vif-expressing Cal51 cells demonstrated a prolonged mitosis of ~16 hr, in contrast to only 30 min in Control cells (*Figure 1B*, *Figure 1—figure supplement 1B*, and *Videos 1–2*). The majority of Vif-expressing cells eventually succumbed to apoptotic cell death or exhibited mitotic slippage, where the cell exited mitosis without completing chromosome segregation (*Figure 1C*).

Vpr has been shown to induce G2/M arrest in a variety of cell types, as evidenced by flow cytometry (*Bartz et al., 1996*; *Elder et al., 2002*; *Emerman, 1996*; *Hall et al., 2024*; *He et al., 1995*; *Jowett et al., 1995*; *Sakai et al., 2006*). We next compared the differences in G2/M arrests induced by Vif and Vpr. While we expected Vpr to induce G2 phase arrest due to its abilities to cause DNA damage, our live-cell imaging revealed that Vpr-expressing cells also experienced a prolonged mitosis of ~1.7 hr, which was significantly shorter than the duration observed in Vif-expressing cells (*Figure 1A–B*). Interestingly, Vif +Vpr expressing cells exhibited a prolonged mitosis lasting ~21.7 hr, indicating that Vif plays a dominant role in mitotic arrest when both proteins are present. Supporting this, the majority of Vif-expressing or Vif +Vpr expressing mitotic cells underwent apoptotic cell death, whereas Vpr-expressing mitotic cells either completed division or experienced mitotic slippage (*Figure 1C*). In summary, although both Vif and Vpr can induce a prolonged mitosis, Vif causes a significantly more severe mitotic arrest, leading to cell death.

To pinpoint the specific sub-stage of mitosis affected by Vif expression, we closely assessed chromosome alignment during metaphase. Notably, most Vif-expressing Cal51 cells successfully achieved metaphase chromosome alignment (metaphase plate) similar to Control cells but with a slight delay, reaching it approximately 1.5 hr post-NEBD compared to the control's ~25 min (*Figure 2A–C*). However, this alignment was unstable and deteriorated over time in Vif-expressing cells. Mitotic arrest induced by common mitotic inhibitors typically occurs in prometaphase, preventing cells from successfully achieving metaphase plate (*Choi et al., 2011*). However, the mitotic arrest caused by Vif was distinctive because cells were able to complete prometaphase but then gradually lost proper chromosome spatial organization over time. Accordingly, we termed this block 'pseudo-metaphase arrest'. Consistent with our findings in Cal51 cells, other commonly used cell lines for cell cycle studies, such as MDA-MB-231 and HeLa, also demonstrated significant mitotic arrest (approximately 12 hr for both) following Vif expression, which subsequently led to either apoptotic cell death or mitotic slippage (*Figure 2—figure supplement 1A–E* and *Figure 2—figure supplement 1F–J*). Similar to Cal51

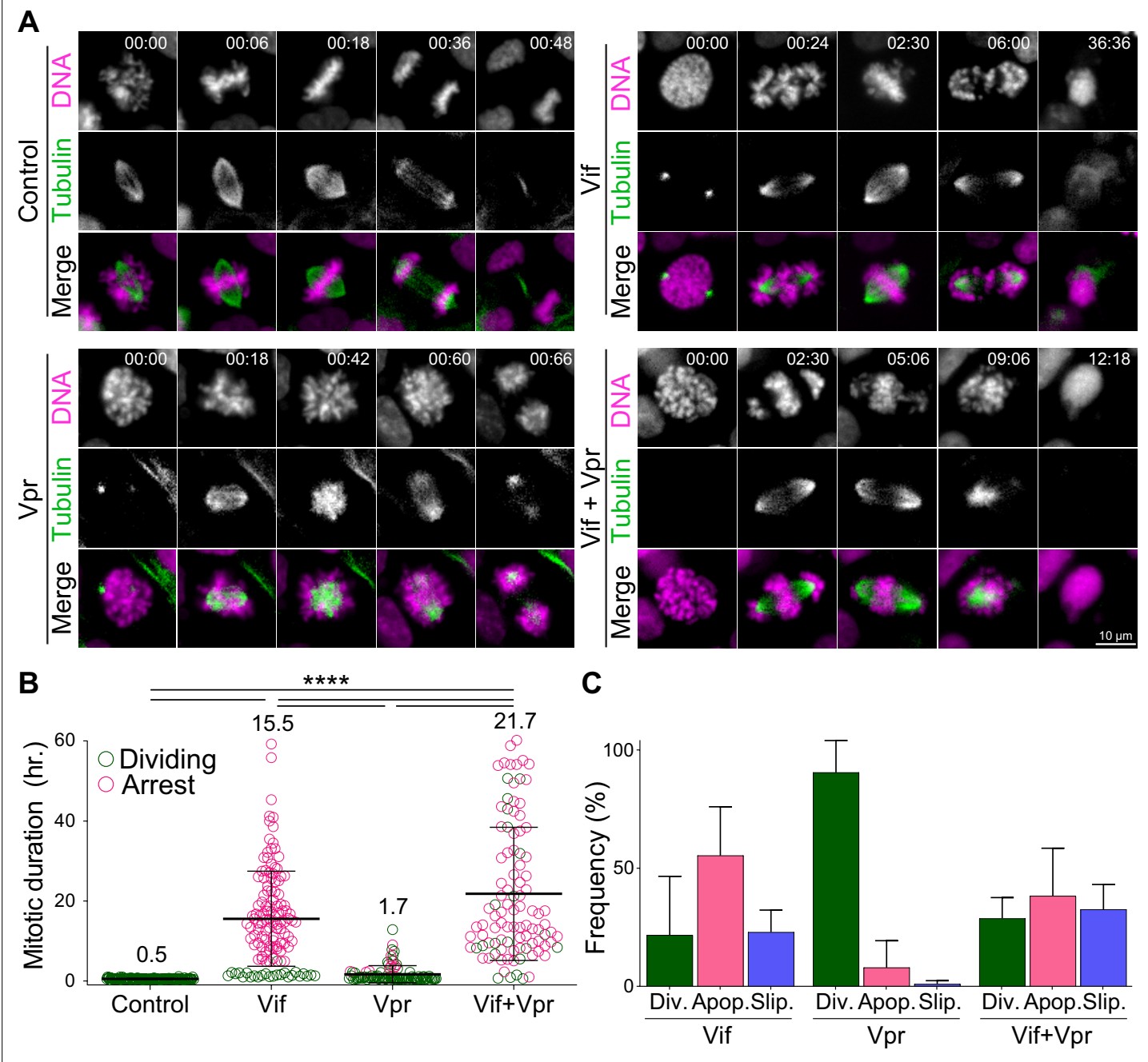

**Figure 1.** Vif and Vpr induce distinct forms of mitotic arrest. (**A**) Representative live cell image for Cal51 cells with H2B-mScarlet and Tubulin-mNeonGreen expressing Control, Vif, Vpr, or Vif +Vpr reporter viruses. (**B**) Average mitotic duration of Cal51 cells expressing respective reporter virus (n=100 for each condition from two replicates). (**C**) Frequency of cell fate after mitosis for Cal51 cells expressing respective reporter virus. (n=100 for each from two replicates).

The online version of this article includes the following figure supplement(s) for figure 1:

**Figure supplement 1.** Vif induces a pseudo-metaphase arrest in Cal51.

cells, the majority of these Vif-expressing cells were able to establish a metaphase plate early but were unable to enter anaphase (*Figure 2—figure supplement 1D–E* and *Figure 2—figure supplement 1I–J*). Consistent with these results, Vif-expressing HeLa cells exhibited a markedly higher mitotic index compared to Control cells at 72 hr post-infection in fixed immunofluorescence (IF) (*Figure 2—figure supplement 1K*). In conclusion, Vif triggers a marked pseudo-metaphase arrest in a range of

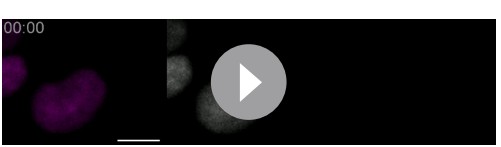

**Video 1.** CFP-positive Control Cal51 live cell imaging. Scale bar represents 10 µm.

https://elifesciences.org/articles/101136/figures#video1

cell lines. Most of these arrested cells experienced either apoptotic cell death or mitotic slippage, suggesting a conserved underlying mechanism.

## Solo Vif expression is sufficient to trigger a robust pseudo-metaphase arrest

To determine if Vif expression alone is sufficient to induce a robust pseudo-metaphase arrest in the absence of other viral factors, we next engineered HeLa cells to conditionally express codon optimized Vif (CO-Vif) under the control of a doxycycline-inducible promoter (*Das et al., 2016*). As a control, we employed the same system but with mNeonGreen expression instead of Vif. Control cells displayed mNeonGreen signals approximately 10 hr post-doxycycline induction. In line with these expression kinetics, cells expressing CO-Vif almost invariably exhibited pseudo-metaphase arrest roughly 10 hr post-induction; with cells arrested for ~15 hr, in contrast to Control cells that completed mitosis in ~1 hr (*Figure 2D–E* and *Videos 3–4*). While Control cells continued to propagate, cells expressing Vif did not, confirming that Vif expression alone is sufficient to trigger prolonged pseudo-metaphase arrest and subsequent apoptotic cell death (*Figure 2F–G*).

## Vif accelerates G2 progression with no effect on the G1 or S phases

We next asked if Vif altered other stages of the cell cycle in addition to mitosis. To this end, we developed a novel method that allowed us to accurately distinguish between G1, S, and G2 phases in individual Cal51 reporter cells during live-cell imaging based on tracking changes to the intensity of Histone H2B-mScarlet over time (see Methods). This method offers the advantage of allowing us to measure temporal changes of the DNA content at single cell resolution with high accuracy. Briefly, during S phase, H2B-mScarlet signals increased steadily, eventually plateauing and remaining constant throughout the G2 phase. *Figure 3A* presents example images and an intensity profile covering the period from the end of one mitosis to the beginning of the next in a Control Cal51 cell. Using this method, we observed no significant differences in the durations of either G1 or S phases between Control and Vif-expressing cells. However, Vif-expressing cells exhibited a slight yet statistically significant reduction in G2 phase duration compared to Control cells (*Figure 3B–C* and *Figure 3—figure supplement 1A*). Consistent to Cal51 cells, Vif expression also did not significantly impact the duration of interphase in two additional cell lines, RPE1 and MDA-MB-231 cells (*Figure 3—figure supplement 1B*). In summary, these findings demonstrated that Vif expression induces pseudo-metaphase arrest without notably affecting the overall duration of interphase (the cumulative time of G1, S, and G2 phases).

## Vif induces pseudo-metaphase arrest independently of p53

A previous study indicated that Vif-induced cell cycle arrest is due to interactions with tumor suppressor p53 (*Izumi et al., 2010*), which is well known for triggering G2 cell cycle arrest in response to DNA damage (*Clair et al., 2004*; *Stark and Taylor, 2006*; *Taylor and Stark, 2001*). Considering that we had already observed Vif inducing pseudo-metaphase arrest in cell lines with functionally inactivated p53, such as MDA-MB-231 (*Olivier et al., 2002*) and HeLa (*Evans et al., 2018*; *Wang et al., 2011*: *Figure 2—figure supplement 1A–E* and *Figure 2—figure supplement 1F–J*), we further investigated the potential p53-dependency by assessing Vif's effects in p53 null knockout (p53 KO) RPE1 (*Mardin et al., 2015*) or HCT116 cell lines (*Figure 3D–G* and *Figure 3—figure supplement 1C–H*). Both wild-type (p53 +/+) and p53 KO RPE1 and HCT116 cells demonstrated significant pseudo-metaphase arrest in response to viral Vif expression. Specifically, RPE1 wild-type cells were arrested for >10 hr, RPE1 p53 KO cells for ~25 hr, and both HCT116 wild-type and p53 KO cells for >6 hr. In contrast, cells infected with

**Video 2.** CFP-positive Vif-expressing Cal51 live cell imaging. Scale bar represents 10 µm.

https://elifesciences.org/articles/101136/figures#video2

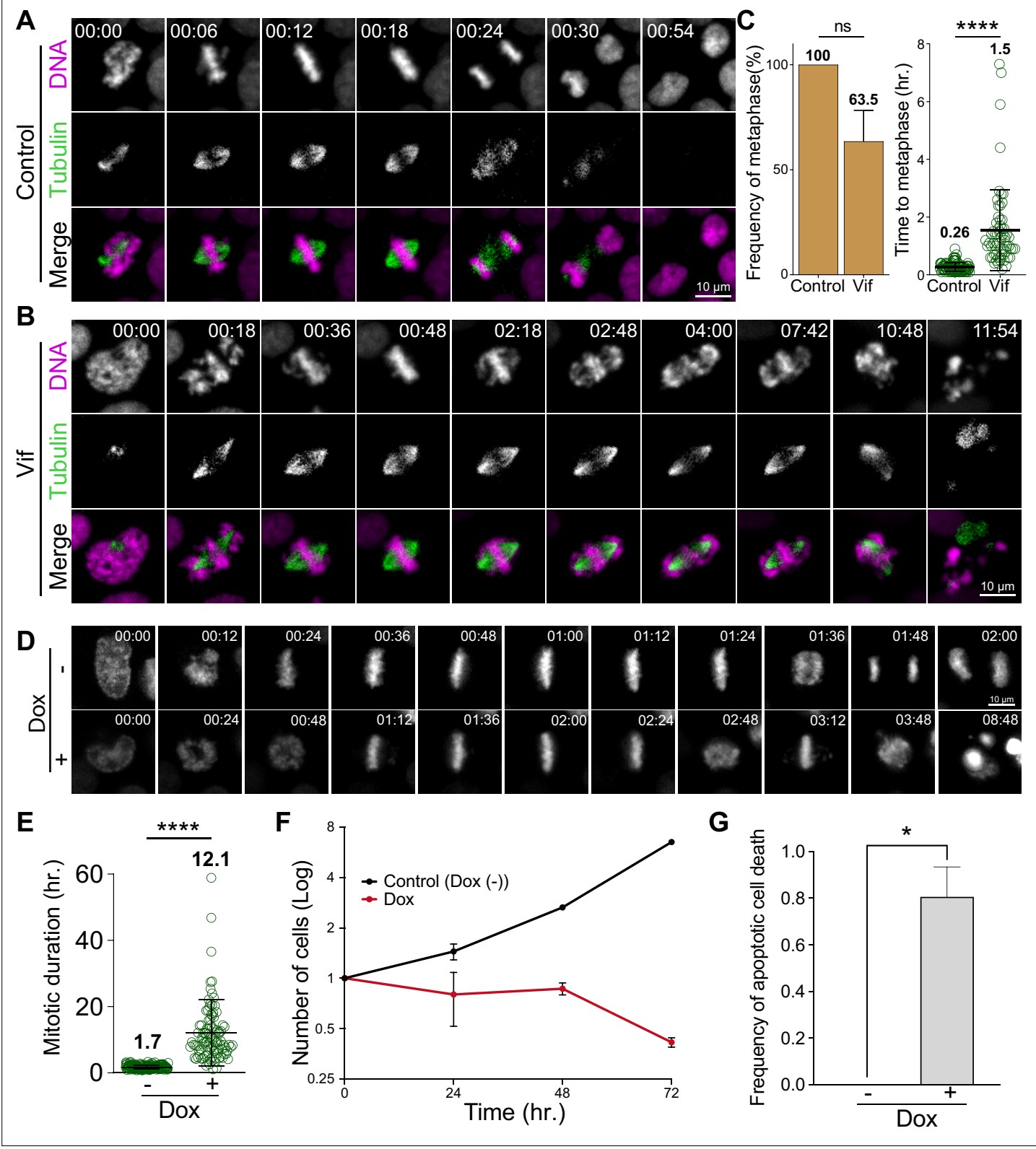

**Figure 2.** Vif induces robust pseudo-metaphase arrest. (**A**) Representative live cell images for Cal51 cells with H2B-mScarlet and Tubulin-mNeonGreen expressing Control reporter virus. (**B**) Representative live cell image for Cal51 cells expressing Vif reporter virus. (**C**) Frequency of cells that achieve metaphase plate and time taken to achieve metaphase plate for cells in (**A**) and (**B**) (n=100 for each condition from two replicates). (**D**) Representative live-cell images of Vif conditional expressed HeLa cell with or without Doxycycline (Dox). (**E**) Average mitotic duration in condition (**D**) (n=100 cells

*Figure 2 continued on next page*

*Figure 2 continued*

for each condition from two replicates). (**F**) Quantification of viable cells over time after Dox induction. (**G**) Quantification of apoptotic cells after Dox induction.

The online version of this article includes the following figure supplement(s) for figure 2:

**Figure supplement 1.** Vif induces pseudo-metaphase arrest in MDA-MB-231 and HeLa cells.

the Control virus showed no delay in mitosis (~30 min for both cell lines; *Figure 3D–E* and *Figure 3—figure supplement 1D–E*). All cell lines, regardless of their p53 status, managed to establish a chromosome metaphase plate in the presence of Vif expression (*Figure 3F–G* and *Figure 3—figure supplement 1F–G*). However, most Vif-expressing cells exhibited apoptotic cell death or mitotic slippage (*Figure 3—figure supplement 1C and H*).

## Vif-induced pseudo-metaphase arrest disrupts spatial organization of chromosomes and spindle poles

To further characterize the mitotic defects caused by Vif expression, we carefully assessed Vif's effects on chromosome alignment at the metaphase plate. To this end, we employed super-resolution microscopy and stained for CENP-C, microtubules, and DNA (see Methods). CENP-C was used as a marker for kinetochores, the platform for microtubule attachment on mitotic chromosomes. Our findings revealed that ~100% of Vif-expressing mitotic cells exhibited misaligned chromosomes, with the great majority of these misaligned chromosomes concentrated at spindle poles as polar chromosomes (*Figure 4A–B*, *Figure 4—figure supplement 1A* and *Videos 5–6*).

We next explored the dynamics of chromosome spatial organization in Vif-expressing cells by using live-cell imaging. To do this, we first quantified the proportion of cells exhibiting polar chromosomes at any time point during metaphase/pseudo-metaphase in following cell lines (HeLa, RPE1, MDA-MB-231, and Cal51 cells). Consistent with our fixed-cell analysis, we observed ~100% of Vif-expressing cells exhibiting misaligned polar chromosomes at some time point during prolonged mitosis, in contrast to Control cells, in which misaligned chromosomes were only rarely observed (*Figure 4—figure supplement 1B*). To define the dynamic nature of chromosome movements, we segmented cells into two compartments, polar and equatorial, and then measured the Histone H2B-mScarlet signals within each of these compartments in Cal51 cells over time (*Figure 4C*). Vif-expressing cells exhibited an initial decrease in the frequency of polar chromosomes shortly after NEBD, but this frequency increased significantly during the extended pseudo-metaphase; with pronounced polar chromosomes comprising ~50% of the total DNA. Notably, these misaligned chromosomes continuously oscillated between the poles and the metaphase plate, as shown in *Figure 4D*.

Consistent with abnormal chromosome dynamics, ~25% of HeLa cells expressing Vif exhibited multi-polar spindles (>2 poles) based on fixed cell analysis (*Figure 4—figure supplement 1C*). To corroborate these findings, we used high-temporal resolution live-cell imaging to track and

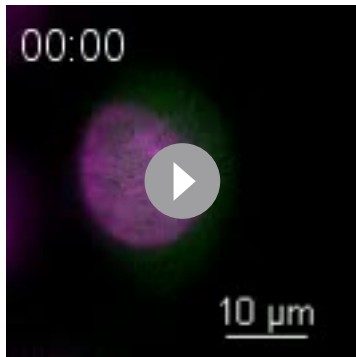

**Video 3.** Tet-on control (mNeonGreen) HeLa live cell imaging.

https://elifesciences.org/articles/101136/figures#video3

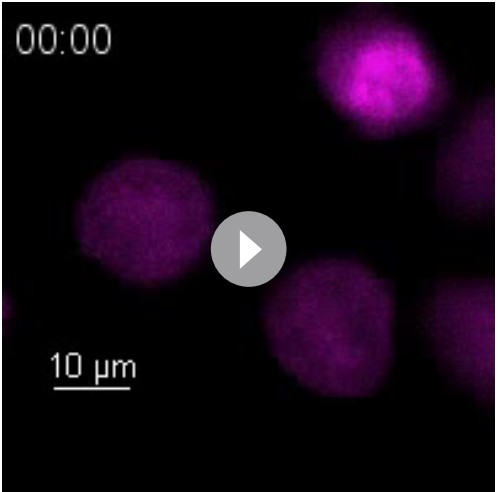

**Video 4.** Tet-on Vif expressing HeLa live cell imaging.

https://elifesciences.org/articles/101136/figures#video4

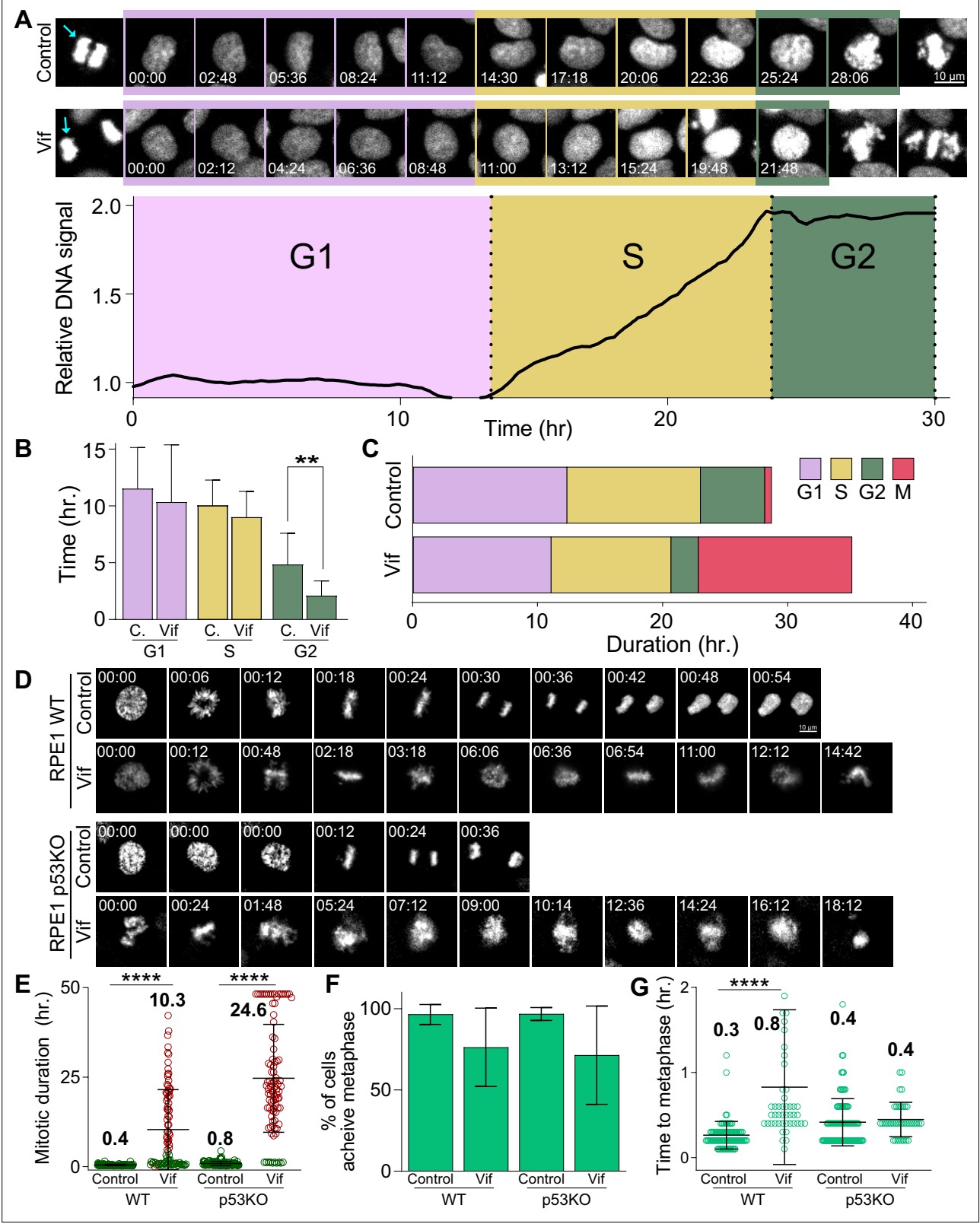

**Figure 3.** Vif does not alter G1 or S phase progression, accelerates G2 progression, and induces pseudo-metaphase arrest independent of p53. (**A**) Top: Representative image of Cal51 cells progressing through G1, S, and G2. Bottom: Representative trace for relative signal intensity of the nucleus through cell cycle. (**B**) Average duration of G1, S, and G2 phases in Control and Vif-expressing cells (n=9 for Control and 11 for Vif, from two replicates). (**C**) Total cell cycle duration for Control and Vif-expressing cells. (**D**) Representative live cell images for Control and Vif-expressing WT or p53 KO RPE1 cells. (**E**)

*Figure 3 continued on next page*

*Figure 3 continued*

Average mitotic duration in WT or p53 KO RPE1 cells (n=>85 cells, from two replicates). (**F**) Frequency of cells which achieve metaphase plate for cells in (**E**). (**G**) Average time taken to achieve metaphase plate for cells in (**F**).

The online version of this article includes the following figure supplement(s) for figure 3:

**Figure supplement 1.** Vif specifically induces pseudo-metaphase arrest independent of p53 status.

quantify spindle poles using mNeonGreen-Tubulin in Cal51 cells. We observed that ~80% of cells expressing Vif demonstrated multi-polarity at some time point during extended mitosis (*Figure 4E*). Moreover, the number of spindle poles varied dramatically in arrested cells, ranging from a monopole to as many as five poles (*Figure 4—figure supplement 1D*).

The integrity of spindle poles is crucial for maintaining the position of the metaphase plate during mitotic progression, so that the length of microtubules making up the mitotic spindle is tightly regulated and typically remains stable until anaphase onset. Interestingly, we found that mitotic spindles in Vif-expressing cells were significantly stretched (~18 µm in length) as compared to Control cells (~12 µm; *Figure 4F* and *Figure 4—figure supplement 1E*). Moreover, although mitotic spindles are typically stationary, we observed spindles in Vif-expressing cells to exhibit dynamic spinning. To define these observations quantitatively, we measured the average angle swept by individual mitotic spindles over time in the presence or absence of Vif expression. We observed a greater than 15-fold increase in the angle covered by spindles in Vif-expressing cells as compared to Control cells (*Figure 4G* and *Figure 4—figure supplement 1F*). In summary, Vif induces dynamic movements in both chromosomes and spindle poles during extended pseudo-metaphase, resulting in severely misaligned polar chromosomes.

## Vif, but not Vpr, disrupts proper localization of PP2A-B56 to kinetochores

Microtubule assembly at the kinetochore is regulated by an intricate network of kinase and phosphatases (*Saurin, 2018*). PP2A-B56 is recruited to kinetochores during prometaphase, where it plays a crucial role in microtubule assembly and the proper alignment of chromosomes (*Foley and Kapoor, 2013*; *Foley et al., 2011*). Previous studies demonstrated that Vif can significantly degrade B56 proteins, as shown in western blots (*Greenwood et al., 2016*; *Marelli et al., 2020*; *Nagata et al., 2020*). Therefore, we asked whether Vif-expressing mitotic cells had diminished B56 at the kinetochores. To investigate this, we performed quantitative immunofluorescence (qIF) using specific antibodies against B56 and CENP-C (as a kinetochore marker) in Control, Vif-expressing, and Vpr-expressing cells. We found that B56 signals at kinetochores, regardless of aligned (equatorial) or unaligned (polar) chromosomes, were significantly reduced in Vif-expressing cells compared to Control and Vpr-expressing cells (*Figure 5A–C*). To determine whether Vif-expressing cells remained free of additional, non-kinetochore-bound pools of B56, we performed qIF in nocodazole-treated cells. It has been demonstrated that nocodazole, a microtubule depolymerizer, can enhance B56 kinetochore localization (*Foley et al., 2011*). As expected, Control cells showed further recruitment of B56 to kinetochores upon nocodazole treatment, whereas Vif-expressing cells did not (*Figure 5A and C*). These results suggest that Vif-mediated degradation of B56 is sufficient to significantly reduce B56 levels at kinetochores during prometaphase, while Vpr has no effect on B56 levels at kinetochores.

To further validate these results, we performed qIF on Polo-like Kinase 1 (Plk1). Plk1 is a key cell cycle regulator, with critical roles at kinetochores for proper mitotic progression (*Colicino and Hehnly, 2018*). It is known that Plk1 levels at kinetochores are regulated by PP2A-B56, and depletion of B56 causes increased levels of Plk1 at kinetochores, leading to improper microtubule attachments (*Foley et al., 2011*). As expected, Plk1 levels at kinetochores were significantly decreased in metaphase as compared to prometaphase in Control cells (*Figure 5D–E*). In Vif-expressing cells, while Plk1 levels at kinetochores on equatorial chromosomes were lower than those on polar chromosomes, Vif-expressing cells showed a global increase in Plk1 levels at kinetochores. More specifically, Plk1 levels at polar chromosomes in Vif-expressing cells were significantly higher than in Control prometaphase, and levels at aligned equatorial chromosomes were also significantly higher than in Control metaphase. In summary, Vif, but not Vpr, diminishes PP2A-B56 levels at kinetochores, resulting in a delay of chromosome alignments.

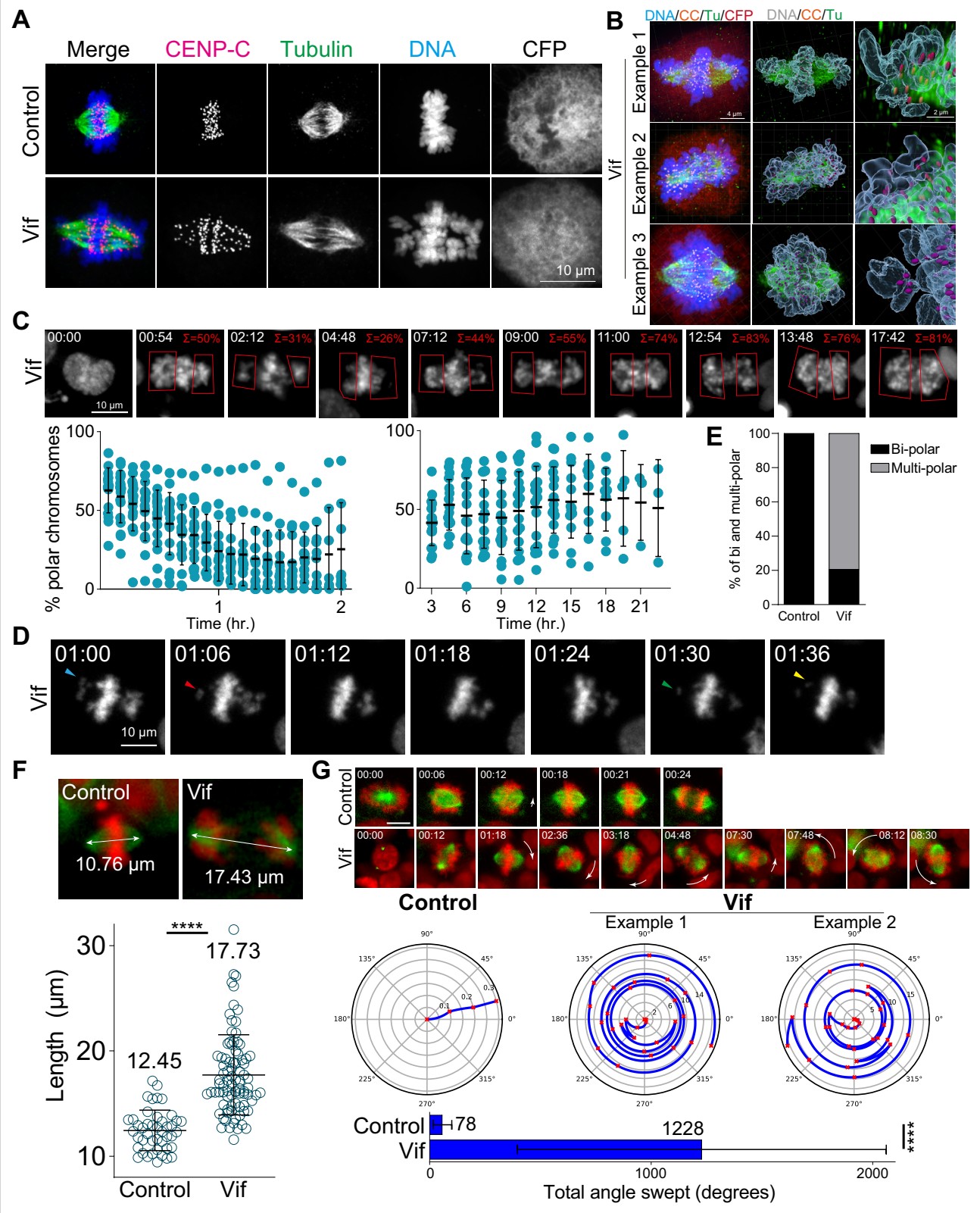

**Figure 4.** Vif induces polar chromosomes, multi-polar spindles, and abnormal chromosome movements. (**A**) Representative immunofluorescence images labeled for CENP-C (as a kinetochore marker), microtubule, and DNA in Control and Vif-expressing HeLa cells. (**B**) Example super-resolution images labeled for CENP-C (CC), microtubule (Tu), CFP, and DNA in Vif-expressed HeLa cells showing polar chromosomes. (**C**) Representative live cell image of Vif-expressing cells where polar chromosomes were quantified by compartmentalizing polar regions. Bottom: Quantification of polar

*Figure 4 continued on next page*

*Figure 4 continued*

chromosome frequency overtime. (**D**) Representative high-temporal live cell images (6 min interval) showing rapid chromosome movement towards and away from the spindle poles. (**E**) Fraction of Cal51 cells showing abnormal number of poles at some point during mitosis. (**F**) Top: Representative images of maximum mitotic spindle length for Control and Vif-expressing Cal51 cells. Bottom: Average maximum mitotic spindle length of Control and Vif-expressing cells. (**G**) Top: Representative live cell image of Control and Vif-expressing Cal51 cells over time showing dynamic spindle spinning. Center: Representative figures showing relative orientation (angle) of the spindle axis over time (radius). Bottom: Average total angle swept during mitosis.

The online version of this article includes the following figure supplement(s) for figure 4:

**Figure supplement 1.** Vif induces unaligned chromosomes, multi-polar spindle, and abnormal chromosome and spindle movements.

## Vif impairs stable and balanced kinetochore microtubule attachments

We demonstrated that Vif-expressing cells exhibited abnormal dynamic chromosome movements (*Figure 4C–D*). Kinetochore-microtubule bindings are cooperatively stabilized by both PP2-B56 and PP1 at kinetochores through an interplay and feedback mechanism (*Saurin, 2018*; *Vallardi et al., 2017*). Consequently, we hypothesized that the reduction of PP2A-B56 by Vif impaired the regulation PP1 phosphatase activities at kinetochores. To test this hypothesis, we quantified the levels of the Astrin-SKAP complex (hereinafter referred to as 'Astrin') at kinetochores by qIF in HeLa cells in the presence or absence of Vif expression. Astrin stabilizes kinetochore-microtubule attachments by recruiting PP1, which dephosphorylates Hec1, a microtubule binding protein at kinetochores, thereby promoting Hec1 binding to microtubules (*Cheeseman et al., 2006*; *Conti et al., 2019*; *Dunsch et al., 2011*; *Manning et al., 2010*; *Schmidt et al., 2010*; *Zhang et al., 2015*). As expected, Astrin signals at kinetochores significantly increased at metaphase compared to prometaphase in Control cells (*Figure 6A–B*). In contrast, Astrin levels at kinetochores on aligned chromosomes (equatorial) in Vif-expressing cells were approximately 50% of control, and Astrin levels on polar chromosomes were largely undetectable (*Figure 6A–B*). We confirmed that levels of CENP-C, which is a core-structural kinetochore protein, did not change between Control and Vif-expressing cells, indicating that the reduction of Astrin in Vif-expressing cells was not due to compromised kinetochore integrity (*Figure 6A–B*).

Generating uniform pulling force across sister kinetochores is essential for maintaining chromosome alignment at the cell equator during metaphase. While control cells showed equal Astrin recruitment at sister kinetochore pairs, consistent with balanced forces (*Figure 6C*), Vif-expressing cells showed significant differences in Astrin levels between sister kinetochores despite CENP-C levels remaining consistent (*Figure 6B–C*).

The N-terminal domain of Hec1 has multiple phosphorylation sites, and dephosphorylation specifically by PP1 is critical for stabilizing its binding to microtubules (*DeLuca et al., 2011*). To directly validate the reduced activity of PP1 at kinetochores in Vif-expressing cells, we performed qIF using a Hec1 phospho-S55 (pS55) antibody (*Figure 6D–F*). As expected, phosphorylation levels (pS55) were high in Control prometaphase and significantly reduced in metaphase (*Figure 6D–E*). In contrast, pS55 levels remained significantly high at aligned chromosomes (equatorial) in Vif-expressing cells

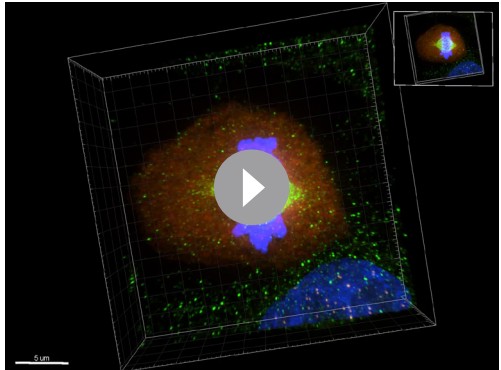

**Video 5.** Super-resolution 3D images of Control HeLa cell.

https://elifesciences.org/articles/101136/figures#video5

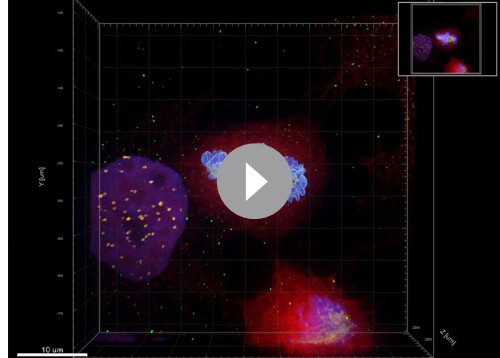

**Video 6.** Super-resolution 3D images of Vif-expressing HeLa cell.

https://elifesciences.org/articles/101136/figures#video6

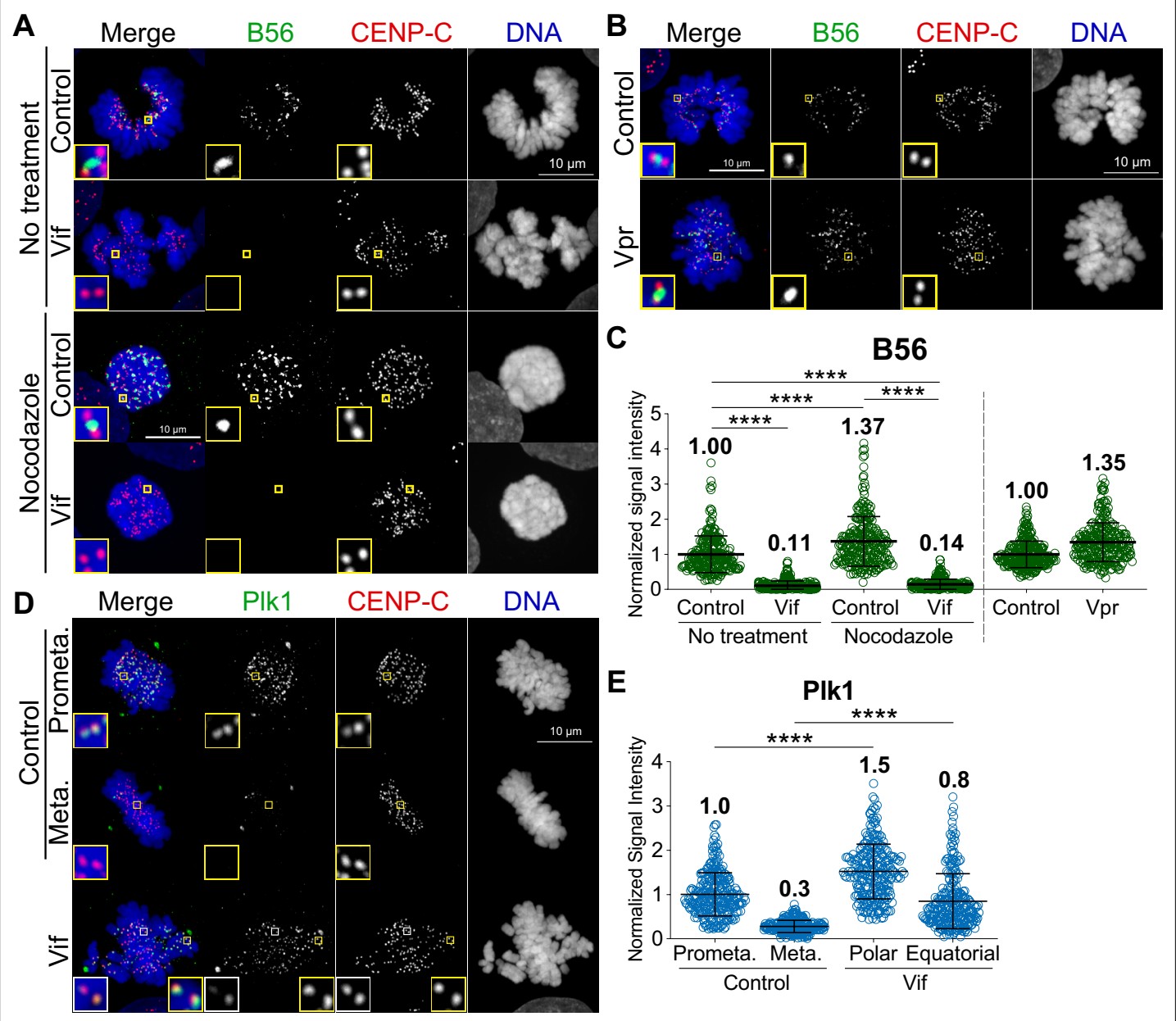

**Figure 5.** Vif, but not Vpr, disrupts the proper localization of PP2A-B56 at the kinetochores. (**A**) Representative immunofluorescence images labeled for B56, CENP-C, and DNA in Control and Vif-expressing HeLa cells with or without nocodazole treatment. (**B**) Representative immunofluorescence images labeled B56, CENP-C, and DNA in Control and Vpr-expressing HeLa cells. (**C**) Normalized B56 intensities at kinetochores for cells in (**A**) and (**B**) (n=200 kinetochores from 8 cells from two independent replicates for each condition). (**D**) Representative immunofluorescence images labeled for Plk1, CENP-C, and DNA of Control and Vif-expressing HeLa cells. (**E**) Normalized Plk1 intensities at kinetochore for cells in (**D**) (n=200 kinetochores from 8 cells from two independent replicates for each condition).

compared to aligned metaphase chromosomes in Control cells. Similarly, unaligned chromosomes (Polar) maintained pS55 levels similar to those in Control prometaphase (**Figure 6D–E**). We confirmed that Hec1 levels at kinetochores were the same in both Vif-expressing and Control cells (**Figure 6G–H**). These results demonstrate that PP1 activity at kinetochores is weaker in Vif-expressing cells compared to Control cells. In agreement with the unbalanced Astrin recruitment between sister kinetochores in Vif-expressing cells, pS55 levels between sister kinetochores were also significantly unbalanced in Vif-expressing cells compared to Control cells (**Figure 6F**). In summary, Vif disrupts the proper assembly of the Astrin-PP1 complex at kinetochores, resulting in the retention of high phosphorylation levels of

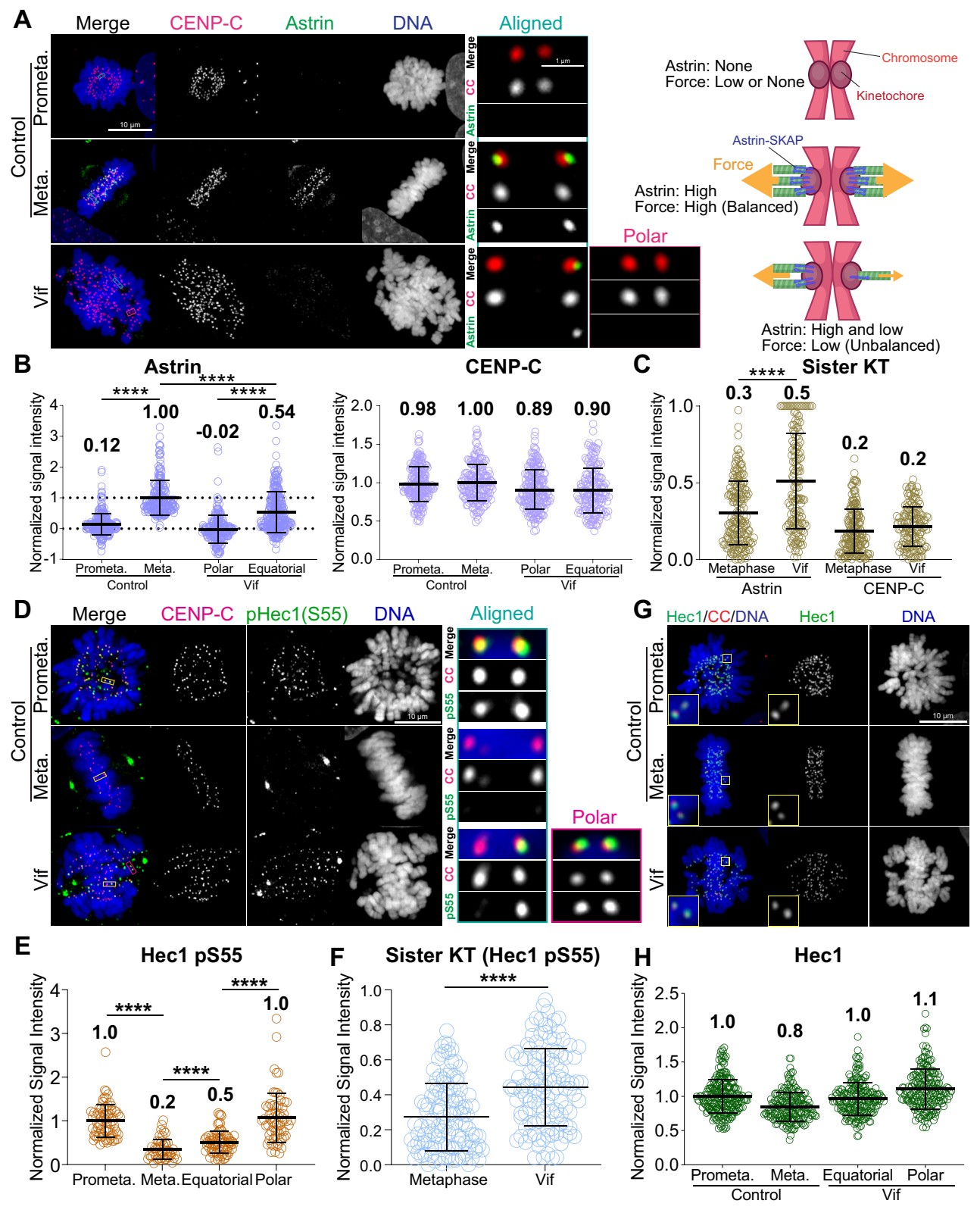

**Figure 6.** Vif impairs stable and balanced kinetochore microtubule attachments. (**A**) Left: Representative immunofluorescence images labeled for CENP-C, Astrin, and DNA in Control and Vif-expressing HeLa cells, Right: Illustrative interpretation of images on the left. (**B**) Normalized Astrin and CENP-C intensities at kinetochores for cells in (**A**) (n=200 kinetochores from 8 cells from two independent replicates for each condition) (**C**) Relative signal intensities of Astrin and CENP-C between sister kinetochores, values normalized with formula: 1 – (lower intensity value/higher intensity value).

*Figure 6 continued on next page*

*Figure 6 continued*

(**D**) Representative immunofluorescence images labeled for CENP-C, pHec1(S55), and DNA in Control and Vif-expressing HeLa cells. (**E**) Normalized pHec1(S55) intensities at kinetochores for cells in (**D**). (n=200 kinetochores from 8 cells from two independent replicates for each condition). (**F**) Relative pHec1(S55) intensities between sister kinetochores, values normalized with formula: 1 – (lower intensity value/higher intensity value). (**G**) Representative immunofluorescence images labeled for Hec1, CENP-C, and DNA in HeLa cells expressing Vif. (**H**) Normalized Hec1 intensities at kinetochores for cells in (**G**) (n=200 kinetochores over 8 cells from two independent replicates for each condition). Representative whole-cell images in (**A**) and (**D**) are maximum intensity projections of multiple z-slices encompassing entire cells, while the zoomed-in images of a single kinetochore pair are presented as either a single z-plane or maximum intensity projections of 2–3 z-slices. This figure was created using BioRender.com.

The online version of this article includes the following figure supplement(s) for figure 6:

**Figure supplement 1.** A protein tag to Vif inhibits Vif's pseudo-metaphase arrest.

Hec1. This leads to weakened and uneven forces between sister kinetochores, likely contributing to dynamic chromosome movements.

## Limitations of the study

To study the spatiotemporal regulation of Vif and the effects of its expression levels at the single-cell level, we aimed to visualize Vif's trafficking during the cell cycle. We discovered that C-terminal fusion of tags, such as 3xHA or mCherry, abolishes Vif's ability to induce pseudo-metaphase arrest (*Figure 6—figure supplement 1A–E*). In this study, we elucidate the mechanisms underlying Vif-induced pseudo-metaphase arrest by utilizing cancer cell lines and a non-transformed normal cell line. While performing similar high-temporal resolution long-term imaging on well-established host cell types for HIV-1 (primary CD4 +T cells, lymphocytes, dendritic cells, or macrophages) poses significant technical challenges, future studies are warranted to investigate these cell types. Such investigations will help determine whether Vif can contribute to the suppression of the host immune system by effectively inducing robust pseudo-metaphase arrest, ultimately leading to cell death.

## Discussion

The specific processes by which HIV-1 causes loss of CD4+ T cells are numerous and include activation of innate immune sensors (*Doitsh and Greene, 2016*), Envelope-driven cell fusion/syncytiation (*Nardacci et al., 2015*), and induction of cell cycle arrest followed by programmed cell death mediated by viral gene products that include Vif and Vpr (*Muthumani et al., 2005*). Vif has recently been shown to induce cell cycle arrest in conjunction with its downregulation of PP2A-B56 (*Marelli et al., 2020*; *Nagata et al., 2020*; *Salamango et al., 2019*). However, the specific nature of this arrest was not previously examined at the single cell resolution and had been assumed to occur during G2 based on flow cytometry assays. In our study, we discovered that expression of Vif actually reduces the duration of G2 (*Figure 3B*) and instead triggers a robust pseudo-metaphase arrest, confirmed in a broad range of cell lines, and with cells typically succumbing to apoptotic cell death after extended pseudo-metaphase (*Figure 1C*, *Figure 2—figure supplement 1C and H*). We also demonstrate that, contrary to a prior study (*Izumi et al., 2010*), Vif-induced pseudo-metaphase arrest occurs independently of p53 status (*Figure 3D–F* and *Figure 3—figure supplement 1D–F*).

Further, we demonstrate that Vif specifically disrupts the kinetochore functions, impairing proper mitotic progression (*Figure 6*). Normally, after NEBD, microtubules efficiently capture kinetochores during prometaphase through the interplay of PP1 and PP2A-B56 phosphatase activities (*Sivakumar and Gorbsky, 2017*; *Smith et al., 2019*; *Vallardi et al., 2017*). PP2A-B56 is recruited to kinetochores in prometaphase, reducing Plk1 activity to facilitate kinetochore-microtubule assembly and promoting recruitment of PP1 by multiple adaptors. A major PP1 adaptor for kinetochore recruitment is the Astrin-SKAP complex whose recruitment requires proper microtubule end-on attachment (*Conti et al., 2019*; *Friese et al., 2016*; *McVey et al., 2021*). In Vif-expressing cells, Vif significantly reduces the level of PP2A-B56 at kinetochore in prometaphase, likely due to its role in B56 degradation. This reduction leads to a slower establishment of metaphase plate (*Figure 7*). The significant loss of PP2A-B56 at kinetochores impairs the feedback control necessary for stabilizing microtubule binding. As a result, there is a significantly lower and uneven recruitment of Astrin-SKAP-PP1 complex to kinetochores, causing uneven pulling forces between sister chromatids that result in some chromosomes being prematurely pulled towards spindle poles preventing metaphase-anaphase transition (*Figure 7*,

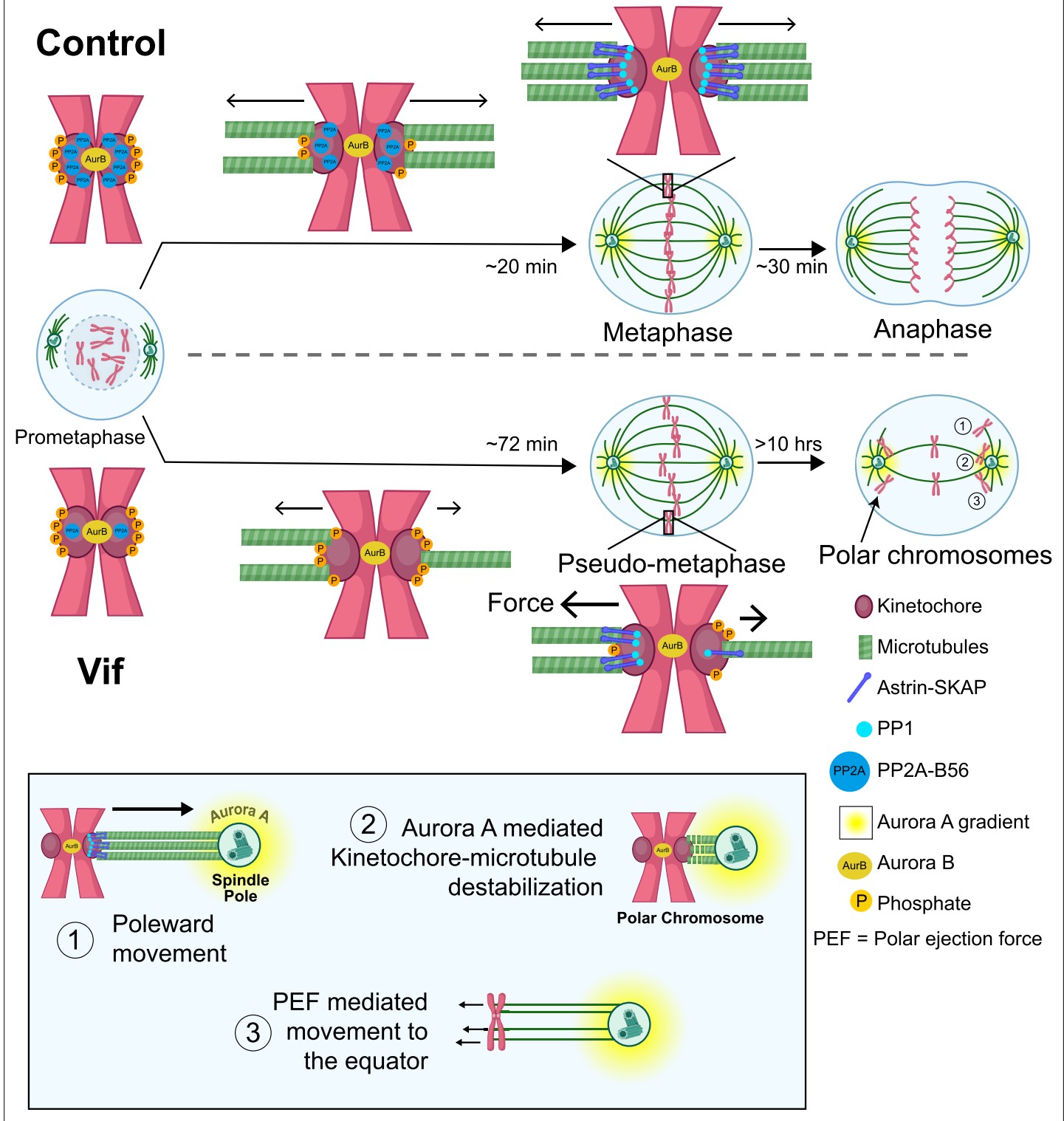

**Figure 7.** Proposed model for the molecular mechanism underlying Vif's pseudo-metaphase arrest. Top: Cartoon model depicting metaphase alignment of Control cells followed by anaphase. Middle: Cartoon model depicting pseudo-metaphase alignment of Vif-expressing cells with unbalanced microtubule attachment followed by three-step polar chromosome cycle. Bottom: Cartoon depiction of three-step polar chromosome cycle, (1) chromosome at the equator is pulled towards a spindle pole due to unbalanced pulling force, (2) kinetochore-microtubule destabilization at the spindle pole, (3) equator-directed movement of chromosome by to polar ejection forces for realignment. This figure was created using BioRender.com.

**bottom, Step 1**). Upon approaching the spindle poles, Aurora A, another key mitotic kinase that regulates mitotic error correction, phosphorylates the microtubule binding domains (MTBDs) of Hec1 and destabilizes kinetochore-microtubule attachment (*Figure 7*, **bottom, Step 2**; *Ye et al., 2015*; *Barr and Gergely, 2007*; *Chmátal et al., 2015*; *DeLuca, 2017*; *DeLuca et al., 2011*; *Kettenbach et al., 2011*). This destabilization of the kinetochore-microtubule attachment could explain why polar chromosomes in Vif-expressing cells lose Astrin signals at kinetochores (*Figure 6A–B*). Polar chromosomes are then transported back to the equator by polar-ejection forces (*Figure 7*, **Step 3**; *Poser et al., 2019*; *Wandke et al., 2012*). The repetition of this cycle accounts for the observed abnormal dynamics of chromosome movements in Vif-expressing cells.

## Methods

### Cell culture

Human HeLa, RPE1, Cal51, and MDA-MB-231 cells were originally obtained from the American Type Culture Collection (ATCC, Manassas, VA, USA). RPE1 p53 KO, HCT116 p53 KO, RPE1 (H2B-RFP), MDA-MB-231 (H2B-mCherry), and Cal51 (Tubulin-mNeonGreen and H2B-mScarlet were endogenously tagged by CRISPR-Cas9) cells were originally obtained from Dr. Jan Korbel, Dr. Yue Xiong (UNC), Dr. Mark Burkard, and Dr. Beth Weaver, respectively. H2B-GFP expressing HeLa cells and conditional CO-Vif-expressing HeLa cells using a pCEP4 vector (Thermo) containing the TRE and Tet promotor with codon-optimized Vif or mNeonGreen were generated in this study. HeLa, MDA-MB-231, HCT116, RPE1 and Cal51 were grown in DMEM high glucose (Cytiva Hyclone; SH 30243.01) or DMEM/F12 (Cytiva Hyclone; SH 3026101) supplemented with 1% penicillin-streptomycin, 1% L-glutamine, and 10% fetal bovine serum under 5% $CO_2$ at 37 °C in an incubator.

### Live cell imaging

RPE1, Cal51, HeLa, and MDA-MB-231 cells were plated on four-chamber 35 mm glass bottom dishes (Cellvis, D35C4-20-1.5-N) or u-Slide 8 well high glass bottom slides (ibidi, 80807) at least 1 day prior to imaging. In a subset of experiments, cells were stained using sirDNA (Cytoskeleton, CY-SC007) for 2 hr prior to imaging to visualize DNA. For conditionally Vif-expressing cells, doxycycline (1 µg/ml, Sigma) was supplemented prior to imaging. High-temporal resolution live-cell imaging was performed using a Nikon Ti2 inverted microscope equipped with a Hamamatsu Fusion camera, spectra-X LED light source (Lumencor), Shiraito PureBox (TokaiHit), and a Plan Apo 20 x objective (NA = 0.75) controlled by Nikon Elements software. Cells were recorded at 37 °C with 5% CO2 in a stage-top incubator using the feedback control function to accurately maintain temperature of growth medium (Tokai Hit, STX model). Images were recorded for 48–120 hr at 6–12 min intervals with three to four z-stack images acquired at steps of 1.5~2 µm for each time point.

### Fixed high- and super-resolution imaging

HeLa cells were fixed using 4% PFA (Sigma) at 24 hr or 72 hr post-infection. Cells were then permeabilized using 0.5% NP40 (Sigma) and incubated with 0.5% BSA (Sigma). Following primary and secondary antibodies were used; CENP-C (MBL), Tubulin (Sigma), GFP (Thermo Fisher), B56-alpha (BD Biosciences), Plk1 (Santa Cruz), Astrin (Sigma), Hec1 (Abcam), Hec1 pS55 (GeneTex), anti-mouse IgG Alexa 488 (JacksonImmuno research), anti-guinea pig IgG Rhodamine Red X (JacksonImmuno research), anti-guinea pig IgG Alexa 647 (JacksonImmuno research), anti-rabbit IgG Alexa-488 (JacksonImmuno research) and anti-rabbit Alexa 647 (Jackson immune research). Stained samples were imaged with either a CSU W1 spinning disc confocal or a CSU W1 SoRa super-resolution (Yokogawa) confocal microscope equipped with a Uniformizer (*Loi et al., 2023*). These spinning disc confocal units were equipped with a Nikon Ti2 inverted microscope with a Hamamatsu Fusion camera, Shiraito PureBox (TokaiHit), and a TIRF SR 100 x objective (NA = 1.49). The microscope system was controlled by Nikon Elements software (Nikon). *Figure 3B* images were generated using Imaris software (Andor).

### Image analysis

Image analysis was performed using Nikon Elements software (Nikon) or Metamorph (Molecular Devices). Mitotic stages and errors were determined by nuclear staining. The mitotic duration was defined as the time from nuclear envelope breakdown (NEBD) to anaphase onset. Timepoints of

formation and loss of metaphase plate were documented. CFP signals were used as a marker for infected cells. Tubulin-mNeonGreen was used for quantifying numbers of spindle poles and monitoring their dynamics. Spindle pole distance was measured when spindle poles were maximally stretched in high-temporal live cell images using Nikon Elements.

## Cell cycle phase analysis

To track cell cycle progression, H2B signals were measured over time using Nikon NIS Elements on time lapse images of Cal51 cells. Signal intensities were measured manually and the local background correction method (*Loi et al., 2023*; *Suzuki et al., 2015*) was applied to accurately quantify chromatin signal intensity. Signals were collected in this manner at 18–30 min intervals. The duration of each cell cycle stage was determined by analyzing changes in the H2B-mScarlet signal over time.

## Polar chromosome quantification

Cell segmentation and measurements of chromosome distribution were performed using the Nikon NIS Elements program. First, the region-of-interest (ROI) tool was used to select chromosomes located at each pole or at the equator. Corrected signal intensity was calculated using a local background correction method (*Loi et al., 2023*; *Suzuki et al., 2015*). Measurements were made for every 6 min for the first 2 hr after NEBD, and for every ~1.5 hr subsequently. For each time point, the percentage of polar chromosomes was calculated using the following formula: (Corrected intensity of Pole1 +Pole2)*100/(Corrected intensity of Pole1 +Pole2+Equator).

## Spindle rotation measurements

Measurements of spindle rotation were performed using the Nikon Elements program's Manual Measurement tool. Cells were observed after NEBD and the free angle tool was used to measure the absolute value of the spindle rotation angle traced from using either the spindle pole or the equatorial chromosomes as a reference. For control cells, measurements were made for each consecutive frame from the first frame where a spindle appeared until the first frame at the onset of anaphase. For Vif-expressing cells, measurements were made for frames whenever a visually significant angle was traced. Data was exported to Excel. The Matplotlib library in Python was used to make polar plots, with time plotted as radius and angle traced plotted as theta.

## Statistics

All experiments were independently repeated two to three times. p-values were calculated using one-way ANOVA and the two-tailed Student's t-test. p-values <0.05 were considered significant. In the figures, p-values are denoted as * for ≤0.05, ** for ≤0.01, *** for ≤0.001 and **** for ≤0.0001.

## Transduction and infection

For infections, growth media was replaced with viral supernatants carrying VSV-G-pseudo typed HIV-1 CFP reporter viruses (Vif-positive or Vif-negative) at a multiplicity of infection of ~1, with the viruses engineered and produced as previously described (*Evans et al., 2018*).

## Acknowledgements

We thank Yuhi Hara, Takanori Tsuchiya, Yoshitaka Sekizawa, Yokogawa Electrical Corporation, Nikon, and Tokai Hit for critical equipment and technical support. We also thank Dr. James Bruce for the critical suggestions and experimental support and Ms. Ainslie Homan for support in data analysis. Illustrations of *Figures 6 and 7* were created using BioRender. Part of this work is supported by the University of Wisconsin-Madison Office of the Vice Chancellor for Research with funding from the Wisconsin Alumni Research Foundation (Research Forward), start-up funding from University of Wisconsin-Madison SMPH, UW Carbone Cancer Center, and McArdle Laboratory for Cancer Research, and NIH grant R35GM147525 and U54AI170660 (to A S) and U54AI170660, R01AI110221, and P01CA022443 (to N S).

## Additional information

### Funding

| Funder | Grant reference number | Author |
|---|---|---|
| University of Wisconsin Office of the Vice Chancellor for Research | Research Forward | Aussie Suzuki |
| National Institute of General Medical Sciences | R35GM147525 | Aussie Suzuki |
| National Institute of Allergy and Infectious Diseases | U54AI170660 | Nathan M Sherer Aussie Suzuki |
| National Institute of Allergy and Infectious Diseases | R01AI110221 | Nathan M Sherer |
| National Cancer Institute | P01CA022443 | Nathan M Sherer |

The funders had no role in study design, data collection and interpretation, or the decision to submit the work for publication.

### Author contributions

Dhaval Ghone, Data curation, Formal analysis, Validation, Investigation, Visualization, Methodology, Writing – original draft, Writing – review and editing; Edward L Evans, Data curation, Formal analysis, Investigation, Methodology, Writing – review and editing; Madison Bandini, Data curation, Formal analysis, Validation, Investigation, Visualization, Writing – review and editing; Kaelyn G Stephenson, Data curation, Formal analysis, Investigation, Writing – review and editing; Nathan M Sherer, Conceptualization, Resources, Supervision, Funding acquisition, Project administration, Writing – review and editing; Aussie Suzuki, Conceptualization, Resources, Data curation, Supervision, Funding acquisition, Visualization, Writing – original draft, Project administration, Writing – review and editing

### Author ORCIDs

Dhaval Ghone ⓘ https://orcid.org/0009-0003-8079-7063
Nathan M Sherer ⓘ https://orcid.org/0000-0001-9974-236X
Aussie Suzuki ⓘ https://orcid.org/0000-0001-7390-5116

Reviewer #1 (Public review): https://doi.org/10.7554/eLife.101136.3.sa1
Reviewer #2 (Public review): https://doi.org/10.7554/eLife.101136.3.sa2
Author response https://doi.org/10.7554/eLife.101136.3.sa3

## Additional files

### Supplementary files

MDAR checklist

### Data availability

Source datasets of this study are available at Dryad https://doi.org/10.5061/dryad.k6djh9wj7 and detailed methods can be found in the figures and the Methods section. Materials used in this study are also available from the corresponding author (A Suzuki) upon reasonable request.

The following dataset was generated:

| Author(s) | Year | Dataset title | Dataset URL | Database and Identifier |
|---|---|---|---|---|
| Suzuki A, Sherer N | 2025 | Ghone et al (2025) HIV-1 Vif disrupts phosphatase feedback regulation at the kinetochore, leading to a pronounced pseudo-metaphase arrest | https://doi.org/10.5061/dryad.k6djh9wj7 | Dryad Digital Repository, 10.5061/dryad.k6djh9wj7 |

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
