## [Editor Report · eLife Assessment]

This study provides a **convincing** explanation for why HIV-1 Vif causes a qualitatively different cell cycle arrest to its accessory gene counterpart Vpr. The authors use elegant time-dependent microscopy reporter assays in immortalized tumor cell models to show that HIV-1 Vif causes a pseudo-metaphase arrest rather than a G2 arrest. The metaphase arrest correlates with dysregulation of the kinetochore that could be explained by the loss of phosphatase functions that determine chromosome-microtubule interactions. These **valuable** findings lay the groundwork for additional studies examining the mechanisms and consequences of this Vif-dependent phenotype in the viral life cycle and in primary cells more relevant to HIV-1 pathogenesis.

---

## [Referee Report · Reviewer #1 (Public review)]

Summary:

Ghone et al show that HIV-1 Vif causes a pseudo-metaphase arrest rather than a G2 arrest. The metaphase arrest correlates with misregulation of the kinetochore that could be explained by the loss of phosphatase functions that determine chromosome-microtubule interactions.

Strengths:

The single-cell imaging using different reporters of cell cycle progression is very elegant and the quantitation is convincing. The authors clearly show that what others have characterized as a G2 arrest by flow cytometry is somewhat later in metaphase and correlates with kinetocore misregulation.

Weaknesses:

(1) The major problem with the paper is trying to connect what is observed in tumor cell lines with actual infections in primary T cells. While all of the descriptive work in cell lines is convincing, none of these cells are relevant targets and tumor cells have different cell death and cell cycle regulation than primary T cells. Thus, while Vif might well do all of the things described in the manuscript, it is a stretch to connect any of it to what happens in vivo. In the revised version, the authors now acknowledge this caveat.

(2) Line 109 and elsewhere. The ability of Vif to cause cell cycle arrest and bind PP2A subunits is not a completely conserved feature. Rather, it is quite variable in different HIV-1 strains. (e.g. https://doi.org/10.1016/j.bbrc.2020.04.123 and https://elifesciences.org/articles/53036). Therefore, it is necessary for the authors to quite clearly use strain designations in the manuscript rather than a generic "Vif", and to more clearly describe the viruses being used. In the revised version, the authors now make this more clear.

(3) Figure 5: This figure shows disruption of PP2A-B56 at the kinetochores. However, is this specific to the kinetochores? Since Vif has been described to more broadly degrade PP2A-B56, could this not be a result of a more general decrease in PP2A activity throughout the cell? In the revised version, the authors now clarify this point.

---

## [Referee Report · Reviewer #2 (Public review)]

Summary

The authors characterize the cell-cycle arrest induced by HIV-1 Vif in infected cells. They show this arrest is not at G2/M as previously thought but during metaphase. They show that the metaphase plate forms normally but progression to anaphase is massively delayed, and chromosome segregation is dysregulated in a manner consistent with impaired assembly of microtubules at the kinetochore. This correlates with the lack of recruitment of B56-subunits of PP2 phosphatase which are known degradation targets of Vif, suggesting that this weakens and unbalances the microtubule-mediated forces on the separating chromosomes.

Strengths

The authors present a very well-performed set of quantitative live cell imaging experiments that convincingly show a difference between Vif and Vpr-mediated cell cycle arrests. Through an in-depth characterization of the Vif-mediated block in metaphase, they make a strong case for this phenotype being tied to the degradation of PP2-B56 by Vif. Furthermore, it is important that they have performed most of these experiments with virally infected cells, meaning that their observations are observable at relevant viral expression levels of Vif.

Comments on revisions:

The authors have addressed the concerns and have discussed them accordingly. I hope they pursue the in vivo relevance in their future work

---

## [Author Response]

The following is the authors’ response to the original reviews.

**Public Reviews:**

**Reviewer #1 (Public review):**
Summary:Ghone et al show that HIV-1 Vif causes a pseudo-metaphase arrest rather than a G2 arrest. The metaphase arrest correlates with misregulation of the kinetochore which could be explained by the loss of phosphatase functions that determine chromosome-microtubule interactions.Strengths:The single-cell imaging using different reporters of cell cycle progression is very elegant and the quantitation is convincing. The authors clearly show that what others have characterized as a G2 arrest by flow cytometry is somewhat later in metaphase and correlates with kinetochore misregulation.

We sincerely appreciate the reviewer recognizing the quality and precision of our study, particularly our use of long-term live cell imaging combined with single-cell resolution analysis.

Weaknesses:(1) The major problem with the paper is trying to connect what is observed in tumor cell lines with actual infections in primary T cells. While all of the descriptive work in cell lines is convincing, none of these cells are relevant targets and tumor cells have different cell death and cell cycle regulation than primary T cells. Thus, while Vif might well do all of the things described in the manuscript, it is a stretch to connect any of it to what happens in vivo.

We fully agree with this point. It is indeed technically challenging to perform 48-120 hours of live-cell imaging at high magnification at short intervals using primary T cells because of their non-adherent nature. We also agree that Vif’s functions in pseudo-metaphase arrest and the consequent induction of cell death, observed in cancer cells (e.g., Cal51, HeLa, and MDA-MB-231 cell lines) or normal non-transformed epithelial cells (e.g., the RPE1 cell line), may differ in T cells. Further studies and refined approaches will be required to address this important question. We have revised the manuscript to include a discussion of this issue in the section of Limitation of this study.

(2) Line 109 and elsewhere. The ability of Vif to cause cell cycle arrest and bind PP2A subunits is not a completely conserved feature. Rather, it is quite variable in different HIV-1 strains. (e.g. https://doi.org/10.1016/j.bbrc.2020.04.123 and https://elifesciences.org/articles/53036). Therefore, it is necessary for the authors to quite clearly use strain designations in the manuscript rather than a generic "Vif", and to more clearly describe the viruses being used.

Thank you for raising this important point. We utilized the NL4-3 strain in our study and have revised the manuscript to specify this detail. While this study uncovered part of the mechanism by which Vif modulates phosphatase regulation during mitosis, further research is required to elucidate the full mechanism, particularly how this degradation induces a robust pseudo-metaphase arrest.

(3) Figure 5: This figure shows disruption of PP2A-B56 at the kinetochores. However, is this specific to the kinetochores? Since Vif has been described to more broadly degrade PP2A-B56, could this not be a result of a more general decrease in PP2A activity throughout the cell?

Thank you for highlighting this critical point. PP2A is a major serine/threonine phosphatase that regulates numerous essential cell cycle processes. To the best of our knowledge, Vif selectively targets the degradation of the B56 family of PP2A regulatory subunits, without affecting other three B-type subunits or the catalytic core of PP2A itself. During early mitosis, all five members of the B56 family (B56α, B56β, B56γ, B56δ, and B56ε) accumulate at kinetochores and centromeres, where they play critical roles in chromosome alignment. Many PP2A-B56 substrates are also localized to kinetochores and chromosomes during mitosis. Depletion of specific B56 isoforms or introduction of phosphorylation-deficient mutants of PP2A-B56 substrates at kinetochores has been shown to result in mitotic defects, underscoring the crucial roles of PP2A-B56 in regulating kinetochore, centromere, and chromosomal functions during mitosis. Interestingly, we observed no significant cell cycle arrest during G1, S, or G2 phases in Vif-expressing cells. While PP2A-B56 likely has important roles outside of mitosis, Vif-mediated degradation of PP2A-B56 appears to selectively disrupt its mitotic functions, particularly at the kinetochore. This finding highlights a targeted mechanism by which Vif interferes with PP2A-B56-mediated regulation of mitotic processes. However, further experiments are required to elucidate the precise mechanisms underlying Vif's inhibition of the specific mitotic roles of PP2A-B56.

**Reviewer #2 (Public review):**
SummaryThe authors characterize the cell-cycle arrest induced by HIV-1 Vif in infected cells. They show this arrest is not at G2/M as previously thought but during metaphase. They show that the metaphase plate forms normally but progression to anaphase is massively delayed, and chromosome segregation is dysregulated in a manner consistent with impaired assembly of microtubules at the kinetochore. This correlates with the lack of recruitment of B56-subunits of PP2 phosphatase which are known degradation targets of Vif, suggesting that this weakens and unbalances the microtubule-mediated forces on the separating chromosomes.StrengthsThe authors present a very well-performed set of quantitative live cell imaging experiments that convincingly show a difference between Vif and Vpr-mediated cell cycle arrests. Through an in-depth characterization of the Vif-mediated block in metaphase, they make a strong case for this phenotype being tied to the degradation of PP2-B56 by Vif. Furthermore, it is important that they have performed most of these experiments with virally infected cells, meaning that their observations are observable at relevant viral expression levels of Vif.

We appreciate the reviewer’s recognition of the importance and significance of our study.

WeaknessesExperimentally there is very little to criticize with respect to the cellular systems used. Data from 10.1016/j.bbrc.2020.04.123 has identified selective mutants that fail to degrade B56 while maintaining A3G degradation by Cul5, and it would be nice to confirm that such a mutant behaves like the delta-Vif virus when examining metaphase, but selective ablation of B56 during mitosis to mimic Vif is would expect to be very challenging and beyond the scope.

Thank you for your valuable suggestion. As also highlighted by Reviewer #1, it is true that certain variants of Vif, as discussed in 10.1016/j.bbrc.2020.04.123, differentially impact B56 degradation. Notably, some variants degrade A3G without inducing cell cycle arrest. We agree that investigating whether Vif's effects on B56 are directly linked to the mitotic arrest phenotype is an important direction for future research. Equipped with our advanced imaging tools, we are now preparing to extend our studies to include Vif variants from additional HIV-1 subtypes, including primary isolates. As you rightly pointed out, depletion of B56 is expected to be challenging as the B56 family comprises multiple isoforms, each with distinct and partially redundant roles in mitosis, particularly in microtubule assembly and spindle assembly checkpoint regulation. The functions of PP2A-B56 in mitosis are well-documented compared to the relatively new studies on Vif’s role in PP2A-B56 degradation. In human cells, the B56 family comprises 5 isoforms (B56α, B56β, B56γ, B56δ, and B56ε). While all B56 isoforms localize to kinetochores or centromeres during early mitosis, the reasons for their slightly different localization patterns (to either kinetochores or centromeres) remain unclear (Vallardi et al., eLife, 2019). Notably, these isoforms exhibit functional redundancy; thus, the depletion of any single isoform does not result in severe mitotic defects (Foley et al., Nature Cell Biology, 2011; Neumann et al., Nature, 2010). Supporting this redundancy, the overexpression of a single isoform (tested only B56α and B56γ) can rescue kinetochore function when all other isoforms are depleted (Foley et al., Nature Cell Biology, 2011; Vallardi et al., eLife, 2019). This complexity poses significant challenges to modulating the relative levels of individual B56 isoforms experimentally. While these specific experiments are beyond the current scope of our study, we remain committed to advancing our understanding of the mechanisms driving Vif-induced pseudo-metaphase arrest. Your suggestion aligns with our ongoing efforts, and we will consider these experiments as we further explore this fascinating area.

Where I would raise some criticism is in the relevance of these observations to the replication and pathogenesis of the virus itself, which the authors do not address or discuss. Firstly, despite clear data that both Vpr and Vif can lead to a cell cycle arrest in cycling cells, it has never been particularly clear why the virus does this. While I would agree with the authors that Vif results in the metaphase arrest through targeting B56-PP2A, this may not be the reason WHY the virus targets one of the cell's major phosphatases, but rather a knock-on effect of doing so. I appreciate that this is beyond the scope of the study, but it is something I feel should be discussed rather than the narrow mechanistic points made in the discussion. Secondly, the authors suggest that this activity of Vif is a major cause of apoptosis in infected cells and perhaps CD4+ T cell depletion in vivo. It would be good to quantify how much apoptosis is Vif-dependent in infected primary human CD4+ T cells rather than transformed tumor cells, and whether this correlates with the Vif-mediated induction of a pseudometaphase.

Thank you for highlighting this important point. We completely agree that the full scope of Vif’s bi-functional roles, in both degrading the APOBEC3 family, which is essential for HIV-1 infection, and inducing cell cycle arrest, is not yet fully understood. The connection between Vif’s role in cell cycle arrest and the HIV-1 life cycle remains unclear. One possible explanation, as discussed in our study, is that Vif-induced pseudo-metaphase arrest may contribute to cell death, suggesting that Vif could play a role in the reduction of CD4+ T cells. Alternatively, Vif’s impact on cell cycle arrest, or its disruption of phosphatase activity, could facilitate HIV-1 virus production. However, further experiments, especially using primary human CD4+ T cells with similar approaches as in this study, are essential to gain deeper insights. This discussion has been included in the Limitations section of our study.

**Recommendations for the authors:**

**Reviewer #1 (Recommendations for the authors):**
(1) The first paragraph of the Introduction is not necessary and anyway is quite outdated about the current state of HIV pathogenesis. Likewise, the discussion implies that HIV pathogenesis is due to virally-induced cell death, which is also outdated by more than a decade of work demonstrating that chronic immune activation is the driver of CD4 cell decline rather than direct cytotoxicity due to viral proteins.

We have revised the first paragraph of the Introduction.

(2) Line 134. I do not know what are Cal51 cells, and why they are being used for an HIV study here. Some rationale for being the cell of choice for this study should be included.

Thank you for this suggestion. We have revised the text to clearly articulate the rationale for selecting the Cal51 cell line in this study. Briefly, this study focuses on the robust mitotic arrest induced by Vif. To capture this phenomenon, long-term live-cell imaging was required with a range of 48–120 hours, with imaging intervals of 6–12 minutes and 3–4 z-stacks per time point. These parameters presented considerable technical challenges. The Cal51 cell line was chosen as it has been genetically engineered by the CRISPR-Cas9 method to express mScarlet-tagged Histone H2B and mNeonGreen-tagged Tubulin, enabling extended live-cell imaging. Furthermore, the Cal51 cell line exhibits wild-type p53 expression and maintains a stable near-diploid karyotype, making it an ideal model for studying cell cycle progression.

(3) A description of the viruses being used is necessary. Although the authors cite a previous paper, the names in that paper do not exactly match the names used here. I presume that is the NL4.3 strain?

Thank you for raising this important point. We utilized the B type HIV-1 NL4-3 strain in our study and have revised the manuscript to specify this detail.